# DDXPlus: A New Dataset For Automatic Medical Diagnosis

**Arsène Fansi Tchango** [*‡], **Rishab Goel** [*†], **Zhi Wen** [‡], **Julien Martel** [§], **Joumana Ghosn** [‡]

[‡] Mila-Quebec AI Institute

{arsene.fansi.tchango,zhi.wen,joumana.ghosn}@mila.quebec

[†] rgoel0112@gmail.com

[§] Work done while at Dialogue Health Technologies Inc.

julien@datadoc.ca

## Abstract

There has been a rapidly growing interest in Automatic Symptom Detection (ASD) and Automatic Diagnosis (AD) systems in the machine learning research literature, aiming to assist doctors in telemedicine services. These systems are designed to interact with patients, collect evidence about their symptoms and relevant antecedents, and possibly make predictions about the underlying diseases. Doctors would review the interactions, including the evidence and the predictions, collect if necessary additional information from patients, before deciding on next steps. Despite recent progress in this area, an important piece of doctors' interactions with patients is missing in the design of these systems, namely the differential diagnosis. Its absence is largely due to the lack of datasets that include such information for models to train on. In this work, we present a large-scale synthetic dataset of roughly 1.3 million patients that includes a differential diagnosis, along with the ground truth pathology, symptoms and antecedents for each patient. Unlike existing datasets which only contain binary symptoms and antecedents, this dataset also contains categorical and multi-choice symptoms and antecedents useful for efficient data collection. Moreover, some symptoms are organized in a hierarchy, making it possible to design systems able to interact with patients in a logical way. As a proof-of-concept, we extend two existing AD and ASD systems to incorporate the differential diagnosis, and provide empirical evidence that using differentials as training signals is essential for the efficiency of such systems or for helping doctors better understand the reasoning of those systems. The dataset is available at https://figshare.com/articles/dataset/DDXPlus_Dataset/20043374.

## 1 Introduction

In a clinical conversation between a doctor and a patient, the patient usually initiates the discussion by specifying an initial set of symptoms they are experiencing. The doctor then iteratively inquires about additional symptoms and antecedents (describing the patient's medical history and potential risk factors), and considers, throughout the interaction, a differential diagnosis, i.e., a short list of plausible diseases the patient might be suffering from (Henderson et al., 2012; Guyatt et al., 2002; Rhoads et al., 2017), which is refined based on the patient's responses. During this multi-step process, the doctor tries to collect all relevant information to narrow down the differential diagnosis. Once the differential diagnosis is finalized, the doctor can ask the patient to undergo medical exams to eliminate most pathologies included in it and confirm the one(s) the patient is suffering from, or can decide to directly prescribe a treatment to the patient.

---

[*]Equal contribution

36th Conference on Neural Information Processing Systems (NeurIPS 2022) Track on Datasets and Benchmarks.

Aiming to aid doctors in such clinical interactions, there has recently been significant progress in building Automatic Symptom Detection (ASD) and Automatic Diagnosis (AD) systems, using machine learning and Reinforcement Learning (RL) techniques (Wei et al., 2018; Xu et al., 2019; Chen et al., 2022; Zhao et al., 2021; Guan and Baral, 2021; Yuan and Yu, 2021; Liu et al., 2022). These systems are meant to collect symptoms and antecedents relevant to the patient's condition, while minimizing the length of the interaction to improve efficiency and avoid burdening the patient with unnecessary questions. In addition, AD systems are tasked to predict the patient's underlying disease to further aid doctors in deciding appropriate next steps. However, this setting differs from real patient-doctor interactions in an important way, namely the absence of the differential diagnosis. Based on the conversation alone, without further information such as the results of medical exams, doctors tend to consider in their reasoning a differential diagnosis rather than a single pathology (Henderson et al., 2012). In addition to presenting a more comprehensive view of the doctor's opinion on the patient's underlying condition, the differential diagnosis helps account for the uncertainty in the diagnosis, since a patient's antecedents and symptoms can point to multiple pathologies. Moreover, the differential diagnosis can help guide the doctor in determining which questions to ask the patient during their interaction. Thus, considering the differential diagnosis is especially important for better and more efficient evidence collection, for accounting for the uncertainty in the diagnosis, and for building systems that doctors can understand and trust. We believe that the absence of this key ingredient in recent AD/ASD systems is mainly due to the lack of datasets that include such information. The most commonly used public datasets, such as DX (Wei et al., 2018), Muzhi (Xu et al., 2019) and SymCAT (Peng et al., 2018), are all designed for predicting the pathology a patient is experiencing and do not contain differential diagnosis data.

To close this gap and encourage future research that focuses on the differential diagnosis, we introduce DDXPlus, a large-scale synthetic dataset for building AD and ASD systems. This dataset is similar in format to other public datasets such as DX (Wei et al., 2018) and Muzhi (Xu et al., 2019), but differs in several important ways. First, it makes a clear distinction between symptoms and antecedents which are not of equal importance from a doctor's perspective when interacting with patients. Second, it is larger in scale, in terms of the number of patients, as well as the number of represented pathologies, symptoms and antecedents. Third, contrary to existing datasets which only include binary symptoms and antecedents, it also includes categorical and multi-choice symptoms and antecedents useful for efficient evidence collection. Moreover, some symptoms are organized in a hierarchy making it possible to design systems able to interact with patients in a logical way. Finally, each patient is characterized by a differential diagnosis as well as the pathology they are actually suffering from. To the best of our knowledge, this is the first large-scale dataset that includes both ground truth pathologies and differential diagnoses, as well as non-binary symptoms and antecedents. To summarize, we make the following contributions:

- We release a large-scale synthetic benchmark dataset of roughly 1.3 million patients covering 49 pathologies, 110 symptoms and 113 antecedents. The dataset is generated using a proprietary medical knowledge base and a commercial AD system, and contains a mixture of multi-choice, categorical and binary symptoms and antecedents. Importantly, it also contains a differential diagnosis for each patient along with the patient's underlying pathology.
- We extend two existing AD and ASD systems to incorporate the differential diagnosis and show that using this information as a training signal improves their performance or helps doctors understand their reasoning by comparing the collected evidence and the differential.

## 2   Existing datasets and limitations

To build machine learning-based AD or ASD systems that medical doctors can trust, one needs to have access to related patient data, namely, the whole set of symptoms experienced by the patient, the relevant antecedents, the underlying pathology, and lastly the differential diagnosis associated with the experienced symptoms and antecedents. Unfortunately, there is no public dataset with all these characteristics. Existing public datasets of medical records, such as the MIMIC-III dataset (Johnson et al., 2016), often lack symptom-related data and are therefore inappropriate for training AD/ASD models. Other datasets, such as DX (Wei et al., 2018) or Muzhi (Xu et al., 2019), are of small scale, and don't necessarily provide a holistic view of the symptoms and antecedents experienced by patients since they are derived from medical conversations. Indeed, if a symptom is not mentioned in a conversation, there is no way to determine if it was experienced by the underlying patient or not.

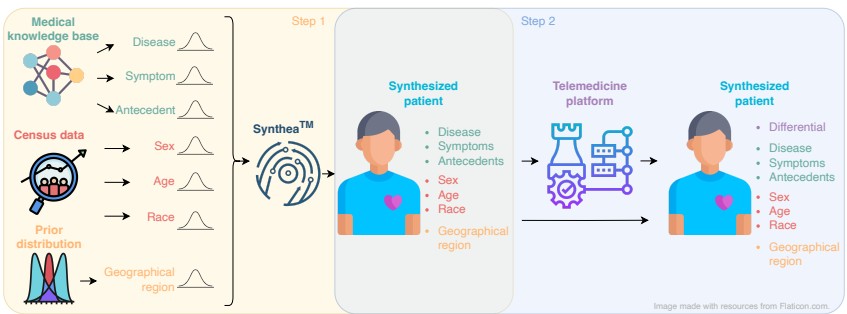

Figure 1: Overview of the data generation process of DDXPlus.

To tackle these limitations, previous works (Peng et al., 2018; Kao et al., 2018) relied on the SymCAT database (AHEAD Research, 2011) for data synthesis. Unfortunately, SymCAT includes binary-only symptoms, which can lead to unnecessarily long interactions with patients, compared to categorical or multi-choice questions that collect the same information in fewer turns. For example, to ask about pain location, systems built using SymCAT need to ask several questions, such as *Do you have pain in your arm?*, *Do you have pain in your ankle?*, $\cdots$, until every relevant location is inquired about. The same information can be obtained by asking a more generic question such as *Where is your pain located?* and obtain all the locations at once. Moreover, as noted by (Yuan and Yu, 2021), the symptom information in SymCAT is incomplete and, as a consequence, the synthetic patients generated using SymCAT are not sufficiently realistic for testing AD and ASD systems. Other datasets with more realistic synthetic patients have recently been proposed based on the Human Phenotype Ontology or HPO (Köhler et al., 2021) and MedlinePlus (Yuan and Yu, 2021). However, like SymCAT, they contain only binary symptoms.

Most importantly, a critical shortcoming that is common to all aforementioned datasets is they do not contain differential diagnosis data. In what follows, we introduce the DDXPlus dataset and describe the undertaken steps towards its creation.

## 3  DDXPlus dataset

To generate the dataset, we proceed in two steps. For each patient, we first use a proprietary medical knowledge base (Section 3.1), public census data, and *Synthea^TM* (Walonoski et al., 2017) to synthesize their socio-demographic data, underlying disease, symptoms and antecedents (Section 3.2). Next, we generate the patient's corresponding differential diagnosis using an existing commercial rule-based AD system (Section 3.3)[2]. Figure 1 illustrates the data generation process.

### 3.1  Medical knowledge base

The dataset we propose relies on a proprietary medical knowledge base (KB) which was constructed by compiling over 20,000 medical papers including epidemiological studies, disease specific articles and meta-analysis papers. From those, the diseases' incidence and prevalence, and symptoms and risk factors' likelihood ratios across various geographies, age and sex groups were extracted. The KB was organized by diseases and reviewed by several doctors, ensuring the diseases' descriptions included atypical presentations. This KB was used to design DXA, a rule-based AD system that has been deployed in a commercial telemedicine platform. In total, the knowledge base covers a set of 440 pathologies and 802 symptoms and antecedents. The pathologies are grouped in overlapping subgroups based on common characteristics referred to as *chief complaints* (Aronsky et al., 2001; Thompson et al., 2006). In this work, we focus on pathologies belonging to the chief complaint related to cough, sore throat, or breathing issues. This group contains 49 pathologies covering 110 symptoms and 113 antecedents. Extending the dataset to all pathologies is left for future work.

Each disease $d$ in the KB is characterized by either an incidence rate, a prevalence rate, or both values. Both rates are conditioned on the age, sex, and geographical region of the patient. The incidence

---

[2]Both the knowledge base and the rule-based AD system used throughout this work are provided by Dialogue Health Technologies Inc. More information about both components is provided in Appendix A

rate measures the proportion of new occurrences of a disease in a population over a period of time while the prevalence rate captures the proportion of individuals in a population that have a disease at a particular time (Friis and Sellers, 2020). For each disease, a set of symptoms and antecedents describing the disease are provided together with their related probabilities. These probabilities are conditioned on the age and sex of the patient. Thus, the probability values $p(s|d, age, sex)$ and $p(a|d, age, sex)$ are provided for each symptom $s$ and each antecedent $a$. In some cases, a symptom $f_s$ (e.g., *pain location*) may be dependent on another symptom $s$ (e.g., presence of *pain*), in which case the KB provides means to extract the corresponding conditional probability $p(f_s|s, d, age, sex)$. Unlike existing datasets mentioned above, evidences (i.e., symptoms and antecedents) within this KB are of several types. They can be binary (e.g., *Are you coughing?*), categorical (e.g., *What is the pain intensity on a scale of 0 to 10?*), or multi-choice (e.g., *Where is your pain located?*). Finally, each disease is characterized by its level of severity ranging from 1 to 5 with the lowest values describing the most severe pathologies.

## 3.2   Patient generation

A synthetic patient is an entity characterized by an *age*, a *sex*, a geographical region *geo*, and who is suffering from a pathology $d$ and is experiencing a set $E$ of symptoms and antecedents. As a first step for modeling patients, we consider the following rule:

$$p(age, sex, geo, d, E) = p(age, sex, geo) \times p(d|age, sex, geo) \times p(E|d, age, sex). \qquad (1)$$

This formulation relies on the fact that the set of evidence $E$ experienced by a patient given a disease doesn't depend on the geographical region. In what follows, we present other rules and assumptions required to exploit the KB.

**Assumptions on the socio-demographic data**   As we only have access to the marginal distributions, we assume that age, sex, and geographical region are independent. That is,

$$p(age, sex, geo) = p(age) \times p(sex) \times p(geo). \qquad (2)$$

This assumption was reviewed by doctors, who deemed it reasonable, as the diseases in the KB, unless specified, are well distributed across populations. The distributions of age and sex can be obtained from census data. For this dataset, we used the 2010-2015 US Census data from the State of New York (US Government, 2015). For more details, see Appendix B. As a result of this choice, the default geographical region of a patient is "North America". Given that some pathologies in the KB can be contracted only if the patient is from a different geographical region, we embed the notion of recent travel when synthesizing a patient. Each synthesized patient is generated by simulating the fact that they recently travelled or not, and if they travelled, in which geographical region. We thus assume the availability of a prior distribution $p(travel)$ representing the proportion of the population travelling each month and we consider that the distribution regarding the geographical regions of destination is uniform[3]. Based on these assumptions, we derive the following prior distribution $p(geo)$ over the geographical regions:

- Sample $u \sim \mathcal{U}(0, 1)$.
- If $u < p(travel)$, then randomly select a geographical region from the available set of geographical regions (see Appendix C). We used $p(travel) = 0.25$ for this dataset.
- If $u \geq p(travel)$, then set the geographical region to be "North America".

**Assumptions on pathologies**   We use the disease's incidence rate, when available, as the disease prior distribution $p(d|age, sex, geo)$, and fall back on the disease's prevalence rate when the incidence rate is missing. This is one of the major limitations of the data generation process which needs to be addressed in future work. The incidence rate can be approximated by dividing the prevalence rate with a constant factor representing the average duration of the disease, and which may be different for each disease. Out of the 49 considered diseases, 8 are affected by this limitation (see Appendix D).

When the resulting rate is greater than 100% (e.g., an incidence rate of 200% means that an individual will likely develop the pathology twice a year on average), we cap it at 100% to avoid having a highly imbalanced dataset. Indeed, without this capping, the dataset would have been dominated by only

---

[3]The dataset can be improved by using real travel and geographical destination statistics.

a few pathologies, with more than half of the patients suffering from the three pathologies whose incidence rate is greater than 100% (i.e., URTI, Viral pharyngitis, and Anemia).

Finally, the KB also contains some diseases that have extremely low incidence rates, and therefore patients suffering from those pathologies are barely generated. To increase the chance of those pathologies to be represented within the dataset, we decide to cap the rate at a minimum of 10%. Thus, the rates used to generate the dataset lie between 10% and 100%. This alteration of the original rates leads to a more balanced dataset (see Figure 2).

**Assumptions on symptoms and antecedents**  Given a sampled disease $d$, the next step is to generate all the evidences (i.e., symptoms and antecedents) the synthesized patient will be experiencing. Given that the KB doesn't contain the joint distribution of all symptoms and antecedents conditioned on the disease, sex and age, a simplifying assumption is made according to which the evidences are independent of each other when conditioned on the disease, age and sex unless explicitly stated otherwise in the KB. In other words, we have:

$$p(E|d, age, sex) = \prod_{e \in E \setminus E_h} p(e|d, age, sex) \prod_{f_s \in E_h} p(f_s|s, d, age, sex), \qquad (3)$$

where $s$ is the symptom which $f_s$ depends on, $E$ is the set of evidences experienced by the patient, and $E_h$ is the subset of symptoms experienced by the patient which are dependent on other symptoms also experienced by the patient.

Some evidences, such as pain intensity, are described as integer values on a scale from 0 to 10. However, the knowledge base only provides the average value of each such evidence given the disease, the age, and the sex of the patient. To inject some randomness in the patient generation process, the values of those evidences are uniformly sampled from the interval $[\max(0, v - 3), \min(10, v + 3)]$ where v is the average value present in the knowledge base.

Finally, in order to reflect the deployed AD system that is based on the KB, we limit to 5 the maximum number of choices associated with multi-choice evidences such as *pain location*.

**Tools**  As mentioned above, we use *Synthea*[TM] to synthesize patients. To generate the value associated with a categorical or multi-choice evidence, *Synthea*[TM] must be provided with a list of possible values together with their related conditional probabilities. *Synthea*[TM] then goes through that list incrementally and decides, for each possible value, if it is "on" or not based on its probability. The process stops as soon as 1 (resp. 5) possible value(s) is (are) "on" for categorical (resp. multi-choice) evidences or the provided list is fully processed. In this work, the possible values of an evidence are ordered in ascending order based on their conditional probability of occurrence to make sure rare values appear in the dataset.

### 3.3  Differential diagnosis generation

As mentioned above, the KB was used to build a rule-based AD system which was deployed in a real-world telemedicine platform and which is capable of generating differential diagnoses. The system was tested in a real world clinical environment, and was vetted by doctors. In the production environment, this platform expects to be given the age, sex and an initial symptom. Based on this information, it determines a set of chief complaints (and their associated pathologies) that are compatible with the provided information, and it engages in a question-answering session with the patient to collect information about the patient's symptoms and antecedents. It then generates a differential diagnosis of ranked pathologies. We leverage this platform to compute the differential diagnosis of each synthesized patient. In order to bypass the limitations of the platform and the errors it might make in the question-answering session by omitting to ask some relevant questions, we provide all evidences in one shot, as opposed to providing each evidence iteratively upon request of the platform. More specifically, we proceed according to the following high-level steps:

- We provide the age and the sex of the patient, the appropriate chief complaint, and we answer "yes" to the question "Are you consulting for a new problem?".
- We provide all the symptoms and antecedents experienced by the patient in one shot, at the beginning of the interaction. The motivation behind this is to make sure that the system is aware of all this information and doesn't miss on any of the patient's symptoms and antecedents.

- The platform may still inquire about additional questions. If that is the case, we answer "no" to those questions until we see a "QUIT" response from the platform or the maximum interaction length (30) is reached.

- When the maximum interaction length is reached, the platform does not produce a differential diagnosis. The corresponding patient is discarded from the dataset.

- When a "QUIT" response is provided by the platform, it contains a differential diagnosis. We further proceed by verifying if the underlying synthesized disease is part of the generated differential diagnosis. If it is not (because the platform itself is not a perfect system or because the patient didn't have enough evidences for the rule-based system to include the simulated disease in the differential diagnosis), the patient is discarded from the dataset. Each pathology within the generated differential diagnosis has a score. Those scores are normalized to obtain a probability distribution.

Given that the platform treats the provided chief complaint as a clue instead of a hard constraint and given that the platform was built to consider all 440 pathologies found in the KB, it sometimes returns a differential diagnosis that contains pathologies which do not belong to the specified chief complaint. There are several options for handling this situation: (1) create an "other pathologies" category and assign it the cumulative mass of the corresponding pathologies, or (2) manually remove those pathologies from the differential diagnosis and re-normalize the distribution. We opt for the second option because we want to restrict the set of pathologies to the universe of 49 pathologies used to synthesize patients. On average, we removed 1.78 ($\pm$1.68) pathologies from the generated differential diagnosis for an average cumulative probability mass of 0.10 ($\pm$0.11). Statistics regarding the rank from which those pathologies are excluded are described in Appendix E.

### 3.4 Dataset characteristics

With the above assumptions and limitations, we generate, under the CC-BY licence, a dataset of roughly 1.3 million patients, where each patient is characterized by their age, sex, geographical region or recent travel history, pathology, symptoms, antecedents, as well as the related differential diagnosis. We divide the dataset into training, validation, and test subsets based on an 80%-10%-10% split, using stratified sampling on the simulated pathology.

Compared to existing datasets from the AD and ASD literature, our dataset has several advantages:

- To the best of our knowledge, this is the first large-scale dataset containing differential diagnoses. This information is important because doctors reason in terms of a differential and not a single pathology, as the evidence collected from a patient can sometimes point to multiple pathologies, some of which requiring additional medical exams before they can be safely ruled out.

- Unlike the *SymCAT* (AHEAD Research, 2011) and the Muzhi (Wei et al., 2018) datasets, which only contain binary evidences, our dataset also includes categorical and multi-choice evidences which can naturally match the kind of questions a doctor would ask a patient, and which can lead to more efficient evidence collection. Moreover, some related evidences are defined according to a hierarchy which can be used to design systems able to interact with patients in a logical way.

- Our dataset makes a clear distinction between antecedents and symptoms.

- Each pathology in our dataset is characterized by a severity level. This information can be used to design solutions that properly handle severe pathologies.

### 3.5 Data analysis

We present summary statistics of the generated dataset. Those statistics are based on the entire set. Statistics on the train, validation, and test subsets are presented in Appendix F.

**Types of evidences**: The distribution of the types of evidences related to the 49 pathologies selected from the KB is provided in Table 1. 6.7% of evidences are categorical and multi-choice.

**Number of evidences**: Table 2 shows an overview of the synthesized patients in terms of the number of simulated evidences. On average, a patient has roughly 10 symptoms and 3 antecedents.

Table 1: Distribution of the evidence types corresponding to the 49 pathologies selected from the KB.

|  | Binary | Categorical | Multi-choice | Total |
|---|---|---|---|---|
| **Evidences** | 208 | 10 | 5 | 223 |
| **Symptoms** | 96 | 9 | 5 | 110 |
| **Antecedents** | 112 | 1 | 0 | 113 |

Table 2: Statistics describing the number of evidences of the synthesized patients.

|  | Avg | Std dev | Min | 1st quartile | Median | 3rd quartile | Max |
|---|---|---|---|---|---|---|---|
| **Evidences** | 13.56 | 5.06 | 1 | 10 | 13 | 17 | 36 |
| **Symptoms** | 10.07 | 4.69 | 1 | 8 | 10 | 12 | 25 |
| **Antecedents** | 3.49 | 2.23 | 0 | 2 | 3 | 5 | 12 |

**Pathology statistics**: Figure 2 shows the histogram of the pathologies of patients in the generated dataset. Although there are three dominating pathologies (URTI, Viral pharyngitis, and Anemia), other pathologies are also well represented.

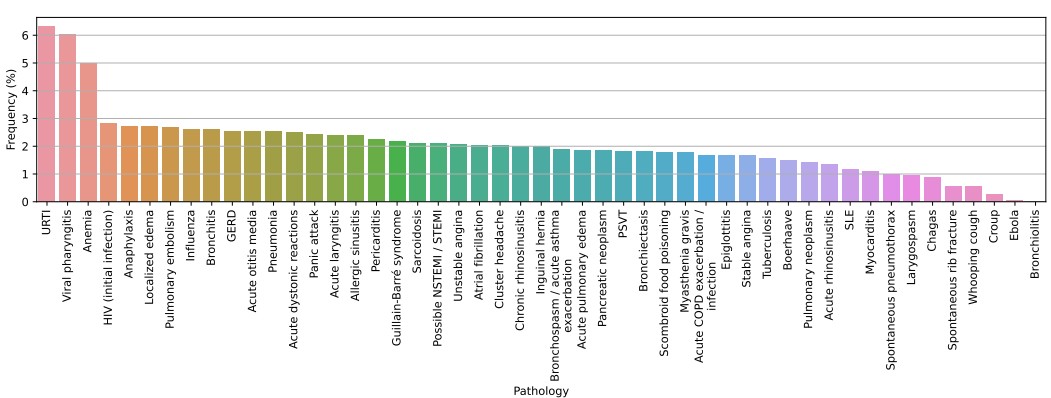

Figure 2: Histogram of the pathologies experienced by the synthesized patients.

**Socio-demographic statistics**: The statistics of the socio-demographic data of the synthesized patients are shown in Figure 3. As expected, these statistics are compliant with the 2015 US Census data of the state of New York used during the generation process (see Appendix B).

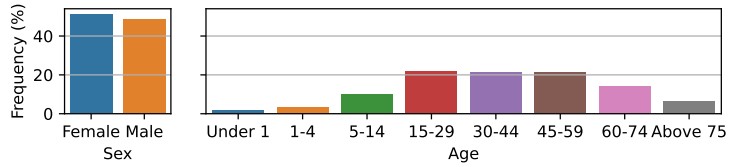

Figure 3: The socio-demographic statistics of the synthesized patients.

**Differential diagnosis statistics**: The distribution of the length of the differential diagnosis of the synthesized patients is depicted in Figure 4 (left). As observed, the generated differential diagnosis can contain more than one pathology. It is also interesting to observe that the simulated pathology is ranked first for more than 70% of patients (see Figure 4 (right)).

**Sample patient**: We present a DDXPlus patient along with a doctor's analysis of this patient and the differential diagnosis. Additional samples are provided in Appendix J.

```
Sex: F, Age: 79
Geographical region: North America
Ground truth pathology: Spontaneous pneumothorax
Symptoms:
---------
        - I have chest pain even at rest.
```

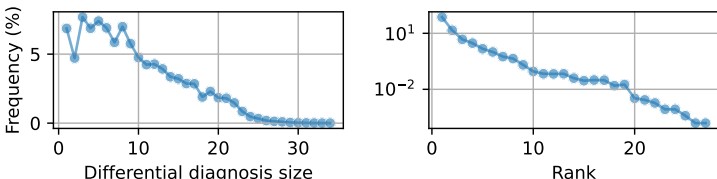

Figure 4: Statistics regarding (left) the length of the differential diagnosis, and (right) the rank of the patient's simulated pathology within the differential diagnosis (y-axis on log scale).

```
        - I feel pain.
        - The pain is:
                * a knife stroke
        - The pain locations are:
                * upper chest
                * breast(R)
                * breast(L)
        - On a scale of 0-10, the pain intensity is 7
        - On a scale of 0-10, the pain's location precision is 4
        - On a scale of 0-10, the pace at which the pain appeared is 9
        - I have symptoms that increase with physical exertion but alleviate with
            rest.
Antecedents:
-----------
        - I have had a spontaneous pneumothorax.
        - I smoke cigarettes.
        - I have a chronic obstructive pulmonary disease.
        - Some family members have had a pneumothorax.
Differential diagnosis:
----------------------
Unstable angina: 0.262, Stable angina: 0.201, Possible NSTEMI / STEMI: 0.160, GERD:
    0.145, Pericarditis: 0.091, Atrial fibrillation: 0.082, Spontaneous
    pneumothorax: 0.060
```

The diagnosis is a spontaneous pneumothorax, which is atypical for this patient's demography. It is rare, but possible. If we were only to collect the positive evidence and score solely on the true or false label, a doctor would find the history missing key questions for a 79 year old female presenting with chest pain that increases with exertion. We definitely need to cover the cardiovascular review more thoroughly and explore the most specific symptoms and risk factors. Optimizing for an accurate differential forces a model to ask questions that in that case would be answered "no", but would have a really positive impact on the confidence of doctors and medical staff in the model's capability to adequately collect a medical history. Without a differential, much would be left out of the medical history.

## 3.6 Dataset usage

We provide a description of the released dataset in Appendix I. Based on the previous sections, it should be clear that the released dataset is meant for research purposes. Any model trained on this dataset should not be directly used in a real-world system prior to performing rigorous evaluations of the model performance and verifying that the system has proper coverage and representation of the population that it will interact with.

# 4 Experiments

## 4.1 Models

We consider two existing AD/ASD systems which were originally designed to predict the pathology a patient is suffering from, and adapt those systems to instead predict the differential diagnosis.

**AARLC** AARLC (Yuan and Yu, 2021) is a model that has two branches, an evidence acquisition branch, trained using RL, whose goal is to determine the next evidence to inquire about, and a classifier branch trained in a supervised way to predict the patient's disease. An adaptive approach is used to align the tasks performed by the two branches using the entropy of the distributions predicted by the classifier branch. We use the same settings as those described in Yuan and Yu (2021) but tune the $\nu$ and $\lambda$ parameters together with the learning rate on the validation set. More details about this approach can be found in Appendix G.1.

**BASD** This supervised learning approach builds on top of the ASD module proposed in Luo et al. (2020) (with the exception of the knowledge graph) and adds a classifier network to predict the underlying patient diagnosis at the end of the acquisition process. The agent is made of an MLP with a number $n_h$ of hidden layers of size 2048 which is tuned together with the learning rate on the validation set. More details about this approach can be found in Appendix G.2.

To assess the impact of the differential diagnosis as a training signal, we train two versions of each approach, one focused on predicting the patient's disease and one trained to predict the differential diagnosis. After training, we use the probability distribution predicted by each version at the end of an interaction as the predicted differential diagnosis. The hyper-parameters of each version are tuned independently and the resulting optimal set is used to report model performance.

## 4.2 Experimental setup

Each patient provides their age, sex, and an initial evidence to the model at the beginning of the interaction. The model then iteratively inquires about a symptom or an antecedent, until either all the relevant symptoms and antecedents have been collected or a maximum number of turns $T$ is reached. At the end of the interaction, a differential diagnosis is predicted. We use $T = 30$ in all experiments. Training is performed using a NVIDIA V100 GPU.

## 4.3 Results

An AD system is typically tasked to collect (i) relevant evidences from a patient, (ii) make accurate predictions regarding the patient's differential, and (iii) operate in a minimum number of turns. As such, we report on the interaction length (IL), and evaluate the evidence collection by measuring the recall (PER). We do not measure the evidence precision as it is sometimes necessary to ask negative questions. Additionally, we calculate the recall (DDR), precision (DDP) and F1 score (DDF1) of the differentials. Finally, we report the accuracy of the inclusion of the ground truth pathology (i.e., the pathology a patient was simulated from) in the predicted differential diagnosis (GTPA@1 and GTPA). Note that GTPA@1, which measures the ground truth pathology accuracy when considering the top entry in the differential, is only relevant for models trained to predict the ground truth pathology. It is not relevant for models trained to predict the differential as the ground truth pathology is not necessarily the top entry in the differential. When predicting the differential, the metrics that matter are the DD-based metrics as well as GTPA (to make sure that the ground truth pathology is in the differential). Definitions of these metrics and other details can be found in Appendix H. To compute these metrics, we post-process both the ground truth differentials and the predicted ones to remove pathologies whose mass is less than or equal to 0.01. This threshold is selected to reduce the size of the differentials by removing highly unlikely pathologies.

Table 3 shows the results obtained for the two approaches. Looking at the performance of AARLC which is an AD system, we observe a significant improvement in the differential performance (i.e., the DD-based metrics) when the model is directly trained to predict the differential. This indicates the importance of using the differential as a training signal as the posterior pathology distribution generated by a model trained to predict the ground truth pathology doesn't correspond to the desired differential. We also observe a significant improvement in the recall of the positive evidence. Interaction length increases when predicting the differential as the model collects more evidence. For the BASD model, the collection of positive evidence doesn't improve; this is expected as the disease classifier branch is only enabled at the end of the interaction. But the system's explainability in the form of the differential significantly improves given that doctors can evaluate the alignment between the collected evidence and the differential. This has the potential of increasing the trust of doctors in such systems. Looking at the GPTA-based metrics, models trained to predict the patient's disease exhibit better GTPA@1 scores than the ones trained to predict the differential diagnosis. This is

expected given that the patient's ground truth pathology is not always ranked at the first position in the corresponding differential diagnosis (see Figure 4 (right)). As for the GTPA metric, all approaches do well, even those trained to predict the full differential without knowing what the ground truth pathology is.

We present in Appendix K the sequence of question-answer pairs and the differential generated by each model for the patient introduced in Section 3.5. The behavior of the models is commented by a doctor, including the alignment between the collected evidence and the predicted differential.

The results that we presented should be viewed as initial baselines. The release of DDXPlus opens the possibility for improving the two models described here as well as developing new ideas for ASD and AD systems capable of collecting useful evidence and generating differentials as doctors do.

Table 3: Interaction length, ground truth pathology accuracy, evidence collection, and differential diagnosis metrics of the trained agents as measured on the test set. Diff indicates whether the agent was trained to predict the differential diagnosis (✓) or not (✗). Except IL, all values are expressed in %. Values indicate the average of 3 runs, and numbers in brackets indicate 95% confidence intervals. A higher GPTA@1 is desired for models trained to predict the ground truth pathology, while a higher GTPA is desired for models trained to predict a differential. For all models, higher PER, DDR, DDP and DDF1 is better.

| Method | Diff | IL | GTPA@1 | GTPA | PER | DDR | DDP | DDF1 |
|--------|------|-----|--------|------|-----|-----|-----|------|
| AARLC | ✓ | 25.75 (2.75) | 75.39 (5.53) | 99.92 (0.03) | 54.55 (14.73) | **97.73 (1.21)** | 69.53 (8.51) | **78.24 (6.82)** |
| | ✗ | **6.73 (1.35)** | 99.21 (0.78) | **99.97 (0.01)** | 32.78 (13.92) | 21.96 (0.30) | **99.19 (0.56)** | 31.28 (0.38) |
| BASD | ✓ | 17.86 (0.88) | 67.71 (1.19) | 99.30 (0.27) | 88.18 (1.12) | **85.03 (3.46)** | 88.34 (1.14) | **83.69 (1.57)** |
| | ✗ | 17.99 (3.57) | 97.15 (1.70) | 98.82 (1.03) | 88.45 (5.78) | 21.89 (0.19) | **99.38 (0.07)** | 31.31 (0.29) |

## 5   Conclusion

We release a large-scale benchmark dataset of roughly 1.3 million patients suffering from pathologies that include cough, sore throat or breathing problems as symptoms. The dataset contains binary, categorical and multi-choice symptoms and antecedents. Each patient within the dataset is characterized by their age, sex, geographical region or recent travel history, pathology, symptoms, antecedents, as well as the related differential diagnosis. We extended two existing approaches from the AD/ASD literature (based on RL and non-RL settings) to leverage the differential diagnosis available in the proposed dataset. The obtained results provide empirical evidence that using the differential diagnosis as a training signal enhances the performance of such systems or helps doctors better understand the reasoning of those systems by evaluating the collected evidence, the predicted differential and their alignment. We hope that this dataset will encourage the research community to develop automatic diagnosis systems that can be trusted by medical practitioners as the latter operate with differential diagnoses when interacting with real patients. In constructing this dataset, we guarantee that the differentials are informed by all relevant positive evidences the patients may have. However, while the rule-based AD system which generates the differentials may further inquire about negative evidences, it is not guaranteed that all relevant negative evidences, i.e., evidences that may change the differentials, are considered. Doing so would be much harder practically as the scope of negative evidences is much larger, and we leave this to future work. While extending the above mentioned approaches, we considered all pathologies as equally important. But in general, when establishing a differential diagnosis, medical practitioners likely ask questions to specifically explore and rule out severe pathologies. The proposed dataset has a severity flag associated with each pathology. This leaves room for exploring approaches that better handle severe pathologies. Finally, we would like to emphasize that this dataset should not be used to train and deploy a model prior to performing rigorous evaluations of the model performance and verifying that the system has proper coverage and representation of the population that it will interact with.

## Acknowledgments and Disclosure of Funding

We would like to thank Dialogue Health Technologies Inc. for providing us access to the knowledge base and the rule-based AD system used throughout this work. We would also like to thank Quebec's Ministry of Economy and Innovation and Invest AI for their financial support.

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
