# OpenReview forum: "DDXPlus: A New Dataset For Automatic Medical Diagnosis"
_NeurIPS.cc/2022/Track/Datasets_and_Benchmarks — NeurIPS 2022 Datasets and Benchmarks _

### Official Review · Reviewer_a8KT · 2022-07-20
**DDXPlus: A New Dataset For Automatic Medical Diagnosis**

**Rating:** 5
**Confidence:** 3
**Clarity:** The paper is well written, and the co…

**Strengths:**

1. The clinical scenario proposed in the paper, i.e., differential diagnosis, has been neglected in the previous works while is of great importance in practice.

2. This dataset is quite large, which is feasible for data-driven models.

3. The dataset combines medical knowledge base information and thus is richer in content than previous datasets.

**Weaknesses:**

1. The main weakness of this dataset derives from the fact that it is synthesized based on a series of very strong assumptions, which can lead to severe problems under inappropriate usage. In particular, these assumptions include:

    - Characterizing each disease by an incidence rate and/or a prevalence rate collected from medical papers.
    - Assuming for a given disease, the evidence experienced by a patient does not depend on the geographical region (If there is any reference to support this, please cite it).
    - Assuming the distributions of age, sex, and geographical region are independent.
    - Assuming the proportion of population traveling each month is 0.25 and the distribution regarding the geographical regions of destination is uniform.
    - Caping the chance of diseases with extremely low incidence rates at a minimum of 10%.

    These assumptions raise the following potential risks:

    - The pathogenesis of human beings is quite complex, yet the above pre-settings are too idealistic.
    - The medical knowledge base used in this work was constructed by compiling medical papers. However, there is a gap between academic papers and the practical situation. Also, the distribution of clinical data varies across regions, races, time periods, etc. These factors can significantly affect the generalizability of this dataset.
    - Taking simple hypotheses to generate patient data while only increasing the prevalence of rare diseases will exacerbate the mismatch between the dataset and the real patient data.

    Thus, the performance of ASD/AD systems is hard to soundly evaluate upon this dataset in the case of violating the actual data pattern, even if only for research purposes.

2. Even though using this dataset solely for training and evaluating the performance of data-driven models, there are also several concerns. The main contribution of this dataset is the introduction of differential diagnosis information. However, all the labels and differential diagnoses are auto-generated by a rule-based system. This incurs the problem that a model trained on this dataset will approximate the rule-based system but cannot surpass it. Therefore, it is necessary to adopt strategies to prevent models trained on this dataset from being another implementation of a rule-based system.

3. The importance of recovering differential diagnoses seems not well demonstrated in the experiment section.
    - From the experimental results in Table 3, models without considering differential diagnosis achieve better overall performance, i.e., high-quality predictions (GTPA@k) and shorter interaction length (IL).
    - The prediction objective of Table 4 is to recover the differential diagnosis, which is somewhat confusing. First, the agent trained to predict the differential diagnosis outperforming the one not exposed to such information on this task is not surprising, even a bit tricky. In addition, it would be better to elaborate on the reasons for treating differential diagnosis as a predictive target and using it as a criterion of model performance.

4. More detailed case studies and/or discussions are expected to support the motivation of this paper.

--------------------------------------------------------------------------------------
I will be glad to raise my score if my concerns are well addressed.

**Additional Feedback:**

It would be better to maintain the relevant data preprocessing, model implementation, and evaluation scripts in a code repository, e.g., GitHub.

**Correctness:**

As a synthetic dataset for medical purposes, it is inevitably facing the problems of data bias, low generalizability, etc. Besides, all the labels and differential diagnoses are auto-annotated without the involvement of experts, raising concerns over the usability of the dataset.


**Documentation:**

There is sufficient detail on data collection and organization. The download URL for the dataset and the code of relevant models are well-attached.


**Ethics:**

The dataset may incur potential issues if it is wrongly used or propagated.


**Relation To Prior Work:**

In Section 2, this paper clearly discusses the difference between the presented and the existing datasets.


**Summary And Contributions:**

This paper presents a large-scale synthetic automatic diagnosis dataset, including 1.3 million patient data. It highlights the importance of differential diagnosis during automatic symptom detection and includes underlying pathologies for each patient in the dataset.

In addition, the authors propose that the symptoms and antecedents are of different importance from a doctor's perspective and should be clearly distinguished. Compared with existing datasets, it contains a mixture of multi-choice, categorical and binary evidence.

To demonstrate the usefulness of incorporating the differential diagnosis, the authors implemented and extended two existing AD and ASD systems upon this dataset.

---

> ### Author Response · Authors · 2022-08-19
> **Responses to questions of Reviewer a8KT**
>
> Thank you for the feedback. Given that you had several questions, our response will be split into several openreview comments due to the 5000 character limit per comment.
>
> Please refer to the “General response to all reviewers” section for overall feedback to all reviewers.
>
> *  The main weakness of this dataset derives from the fact that it is synthesized based on a series of very strong assumptions, which can lead to severe problems under inappropriate usage. In particular, these assumptions include:
> Characterizing each disease by an incidence rate and/or a prevalence rate collected from medical papers.
>
> Following is feedback provided by the doctor working with us: ”As the dataset was created mainly to build interactions with agents, it is very appropriate to use incidence and prevalence to produce a diseases distribution. Further more, the incidence rate are sex/age and geographically specific, making it robust. The diseases in the dataset are also mostly uniformly distributed around the world and I do not see any issues with it.”
>
> * Assuming for a given disease, the evidence experienced by a patient does not depend on the geographical region (If there is any reference to support this, please cite it).
>
> Following is feedback provided by the doctor working with us: “The possible evidences vary across countries as endemic diseases can have specific symptoms otherwise not found in other diseases elsewhere. Other than that, no assumption otherwise were made, as the same evidence is experienced in a similar fashion across geographical regions, although the patient description of symptoms can vary widely across cultures.”
>
> * Assuming the distributions of age, sex, and geographical region are independent.
>
> Following is feedback provided by the doctor working with us: “We have to assume they are to account for travel. We focused on the distribution of diseases based on the patient’s demographic, not the opposite. Unless for sex specific diseases, the difference in the diseases distribution across sexes is irrelevant in clinical practice.”
>
> * Assuming the proportion of population traveling each month is 0.25 and the distribution regarding the geographical regions of destination is uniform.
>
> Following is feedback provided by the doctor working with us: “This assumption is not a risk, but rather a safety measure to account for a wider differential.”
>
> * Caping the chance of diseases with extremely low incidence rates at a minimum of 10%.
>
> Since the dataset was built with creating interactions with trainable agents, we want to have enough representation of rarer disease to have a statistical impact, as even some diseases are rare, as a professional, even if the pre test probabilities are low, we still want to inquire about those. 10% was chosen arbitrarily. We rather lose a bit of specificity and gain sensitivity when we evaluate a patient.

---

> > ### Author Response · Authors · 2022-08-19
> > **Additional responses to questions of Reviewer a8KT**
> >
> > * These assumptions raise the following potential risks:
> > The pathogenesis of human beings is quite complex, yet the above pre-settings are too idealistic.
> >
> > Following is feedback provided by the doctor working with us: “The dataset is about the end result of pathogenesis, not the process. The diagnosis entities are actually well defined and studied. The clinical experts were in agreement and satisfied with the disease models produced.”
> >
> > * The medical knowledge base used in this work was constructed by compiling medical papers. However, there is a gap between academic papers and the practical situation. Also, the distribution of clinical data varies across regions, races, time periods, etc. These factors can significantly affect the generalizability of this dataset.
> >
> > Following is feedback provided by the doctor working with us: “That is why doctors reviewed the knowledge base and used their experience to ensure adequate representation of the diseases, especially edge cases and atypical presentation. For the diseases we chose, we don’t expect races, periods and geography to be important aspects, except when indicated in the knowledge base in the antecedents and for endemic diseases specific to geography.”
> >
> > * Taking simple hypotheses to generate patient data while only increasing the prevalence of rare diseases will exacerbate the mismatch between the dataset and the real patient data.
> >
> > We acknowledge the assumptions made for the generation of this dataset. However, we are not claiming that we are releasing a dataset which is perfectly aligned with an epidemiologic situation in all regions. Instead, the primary goal of this dataset is to allow the ML community to propose ideas for AD/ASD systems  using a dataset that is better aligned with elements medical practitioners care about than the other datasets.
> >
> > With that in mind, and given that ML models need examples to learn from, we were OK to increase the prevalence of rare diseases just to obtain a more balanced dataset.  Consequently, an AD/ASD system trained using this dataset would likely be able to deal with patients suffering from these rare diseases even though the number of such patients would be extremely low.
> >
> > As pointed out in the “General response to all reviewers” section, we would like to emphasize that the problem definition as specified by the ML community is incomplete and incorrect due to the lack of collaboration with the medical community and due to the lack of publicly available datasets that are more aligned with the needs of the medical community. DDXPlus is a dataset that can be used to advance the work done by the ML community in a way that is aligned with the problem definition of the medical community. The ML community is still very far from having models that can be applied to real patients. It first needs to re-adjust the problem definition and use datasets that support this definition to gain the trust of the medical community. The doctor with whom we’ve been working  wasn’t concerned by the increase of the rate of rare diseases. He instead wanted to see if it is possible to build ML systems that have the proper behavior when analyzing a patient.
> >
> > * Thus, the performance of ASD/AD systems is hard to soundly evaluate upon this dataset in the case of violating the actual data pattern, even if only for research purposes.
> >
> > The actual data pattern was altered in order to obtain a more balanced dataset that ML models could leverage to learn how to interact with patients suffering from different pathologies even if the proportion of actual patients suffering from those pathologies in the dataset are different from what is observed in real life.

---

> > > ### Author Response · Authors · 2022-08-19
> > > **More responses to questions of Reviewer a8KT**
> > >
> > > * Even though using this dataset solely for training and evaluating the performance of data-driven models, there are also several concerns. The main contribution of this dataset is the introduction of differential diagnosis information. However, all the labels and differential diagnoses are auto-generated by a rule-based system. This incurs the problem that a model trained on this dataset will approximate the rule-based system but cannot surpass it. Therefore, it is necessary to adopt strategies to prevent models trained on this dataset from being another implementation of a rule-based system.
> > >
> > > When we used the rule-based system to generate the differential, we fed all the patient evidence at once to the system to make sure that the system could use all the available information to generate the differential. But in practice, an ASD or AD system has to figure out which evidence to ask about. Those systems cannot replicate the approach we used to generate the differential as they are not fed the patient evidence. Instead, they are responsible for collecting this evidence and then generating the differential. If they have a poor inquiry policy, they will generate poor differentials. Moreover, generating the differential is not the only goal of such systems. To be accepted by doctors, those systems must generate the differential, ask relevant and sufficient questions about evidence, and properly handle severe pathologies (to explicitly rule them in or out of the differential).
> > >
> > > * The importance of recovering differential diagnoses seems not well demonstrated in the experiment section.
> > > From the experimental results in Table 3, models without considering differential diagnosis achieve better overall performance, i.e., high-quality predictions (GTPA@k) and shorter interaction length (IL).
> > >
> > > Predicting the true pathology from the patient evidence is not a medically sound approach, as the evidence is sometimes insufficient to pinpoint the single pathology the patient is suffering from and medical follow-ups are required to identify this pathology. Doctors generate a differential, which is a short list of pathologies that is derived from the patient evidence. To provide a simple example based on the recent covid pandemic: imagine a patient experiencing fever and cough. Based on those symptoms, it is impossible to determine if the patient is suffering from covid, the flu, pneumonia or another relevant disease. Testing is required. For example, the doctor might ask the patient to do a covid test. A model that predicts a single pathology based on this patient evidence is a poor model that will be categorically rejected by the medical community. This is why the GTPA metric is not meaningful. We are presenting it simply because this is the metric used in other AD/ASD papers that do not predict the differential. The metrics that matter are DDF1 and PER. There are additional metrics that matter and haven’t been introduced in this paper: those are metrics related to measuring a model’s ability to properly handle severe pathologies and metrics to quantify whether the sequence of questions asked by a model follows the approach used by doctors. We are working on proposing such metrics based on our collaboration with doctors.
> > >
> > > As for the degradation of the GTPA when using the differential, this is to be expected, in particular for GTPA@1, because the ground truth (GT) pathology is not always the most likely pathology in the differential, as shown in Figures 12, 14 and 16. We would like to reiterate that predicting the ground truth pathology is not a medically sound approach. This is the approach that has been used in the AD/ASD papers published by the ML community but those papers are not solving the problem as defined by the medical community. It is paramount that we work with the medical community to properly define the problem and build useful models.
> > >
> > > Following is additional feedback provided by the doctor about the GTPA being worse when predicting the ddx: “This is expected. In clinical practice, the GT pathology cannot often be verified or is not known initially. For example, patients will be admitted based on suspicion of diseases from a differential produced by a first doctor. Then during the admission, the GT pathology will be discovered from testing, imagery, etc... The ddx is then the only thing of importance in the beginning of the evaluation as it allows to plan further testing and care planning. In other words, even if some evidence is not found, the pertinent negatives and the suspected differential has more value in practice when the presentation of disease is atypical.”

---

> > > > ### Author Response · Authors · 2022-08-19
> > > > **Final responses to questions of Reviewer a8KT**
> > > >
> > > > * The prediction objective of Table 4 is to recover the differential diagnosis, which is somewhat confusing. First, the agent trained to predict the differential diagnosis outperforming the one not exposed to such information on this task is not surprising, even a bit tricky. In addition, it would be better to elaborate on the reasons for treating differential diagnosis as a predictive target and using it as a criterion of model performance.
> > > >
> > > > The differential diagnosis is at the center of a doctor’s reasoning when interacting with a patient. Doctors do not reason in terms of a single pathology because the evidence experienced by a patient is often insufficient to pinpoint the pathology the patient is suffering from. Instead, doctors generate a differential, which corresponds to a set of likely pathologies the patient might be suffering from and that cannot be further refined without additional exams. See the covid example listed above. We would like to reiterate that the current AD/ASD approaches published by the ML community are not solving the right problem. Building models that predict the patient’s ground truth pathology is medically not sound and those models will be categorically rejected by the medical community. It is paramount that the ML community works with medical experts to properly define the problem.
> > > >
> > > > What we wanted to show with the experiments was that extending existing approaches to predict the differential won’t hurt their ability to recover positive evidence. However, this is not sufficient. Predicting the differential is of crucial importance from a medical point of view, and efforts need to be put to design solutions that are more aligned with the needs of medical practitioners.
> > > >
> > > > * More detailed case studies and/or discussions are expected to support the motivation of this paper.
> > > >
> > > > Please refer to the description in the “General response to all reviewers” section. We also added examples of patients in the paper in Sections 3.5 and A.9. And we added an analysis done by the doctor of the predictions made by the models for one patient in Section A.10.
> > > >
> > > >
> > > > * As a synthetic dataset for medical purposes, it is inevitably facing the problems of data bias, low generalizability, etc. Besides, all the labels and differential diagnoses are auto-annotated without the involvement of experts, raising concerns over the usability of the dataset.
> > > >
> > > > Following is feedback provided by the doctor: Although a synthetic dataset is not perfect, it offers many advantages over traditional datasets from medical records. As a medical expert and doctor, I can say that we often under-document encounters and write very brief medical notes that would lack details to train an agent. Synthetic datasets have the advantage of more overall information. The generalizability is in our opinion better with synthetics as the generated patients are not geography bound. Also with the specialization of medicine and institutions, gathering a dataset that has sufficient data coverage would be a great challenge.
> > > >
> > > > A doctor reviewed samples of the synthetic dataset and found that the synthetics patients characteristics represented well the diseases and that the cases covered well typical and atypical presentations.
> > > >
> > > > * The dataset may incur potential issues if it is wrongly used or propagated.
> > > >
> > > > As mentioned in the text, we highly suggest that this dataset should not be used to train and deploy a model prior to performing rigorous evaluations of the model performance and verifying that the system has proper coverage and representation of the population that it will interact with.
> > > >
> > > > *  It would be better to maintain the relevant data preprocessing, model implementation, and evaluation scripts in a code repository, e.g., GitHub.
> > > >
> > > > We do have a GitHub repository (https://github.com/bruzwen/ddxplus) and we will maintain all the suggested information therein.

---

### Official Review · Reviewer_USF5 · 2022-07-22
**Relevant dataset but paper requires further details and experiments**

**Rating:** 5
**Confidence:** 3

**Strengths:**

Besides its size (skating over "quality vs. quantity"), the dataset introduced in the paper provides an important contribution, as it extends the spectrum of existing datasets by providing data for what should be a critical component in ASD/AD systems, namely differential diagnosis.
This dataset can have a positive impact to the community to which it is addressed and will allow to foster further research on AD/ASD systems. Being a synthetic dataset it also presents the benefits of being freely sharable, thus completely removing any barrier to its access and use by other researchers.


**Weaknesses:**

I would like to restate that the authors provide a relevant contribution to the field with the dataset they generated, and that orchestrating all the necessary resources to create the data must have been an important effort.

However, I think that two aspects of the manuscript need further attention, namely: (a) the description of the tools used to generate the dataset and (b) experiments performed.

W.r.t. (a) I cannot help but to notice that while the KB used to generated the data is adequately described in terms of its content, little to no information is provided as to "how" it was compiled. The only reference I could find was "which was constructed by compiling over 20 000 medical papers". Since this is a primary ingredient of the pipeline, I think that further details should be provided, e.g.: exactly who compiled it? Medical data can be "tricky" to interpret, was there any measure of the quality of the KB, e.g. in terms of agreement on diseases/sympots? What kind of medical papers? Where did they come from? I am not sure whether the nature of the resource prevents it, but answers to those questions seems to me mandatory. The same line of reasoning goes to for the "rule-based ASD" system used to generate the differential diagnosis. I think that the reader should be informed at least minimally as to how the system works, e.g. how are the different inputs processed and combined to arrive at the differential diagnosis?

W.r.t. (b), since the authors introduce a large-scale synthetic dataset, my expectation was that one evaluation experiment would have been devoted at cross-dataset validation, i.e. training a system on the synthetic dataset and then validate the gain in performance on a new unseen dataset. This would be a critical testbed considering the type of applications to which the dataset is aimed.
Additionally, I think that experiment section would greatly benefit from an (even minimal) error analysis in order to assess the beneficial impact of the dataset. For instance, I was surprised to see that there wasn't a clear cut improvement in the "Positive evidence" (PE) - R/P/F1 metrics. Intuitively speaking, if a system is trained to predict a differential diagnosis (i.e. multiple diseases), shouldn't the coverage of the symptoms/antecedents be broader? While for AARLC system this seems to be the case (+$\sim$20 in PER) it seems to have no effect for BASD.





**Additional Feedback:**


- line 109:  As a reader, I think I would like to know right away that the KB and the rule-based ASD system used to generate the data where provided by "Dialogue Health Technologies Inc" straight away, e.g. by having a footnote, rather finding this out in the acknowledgements.
- line 158-161: should mention at least a suggestion to overcome the limitation
- line 164-168: if the KB was compiled by medical papers, the "domination" of specific pathologies is "reflecting" the true data distribution. Why change it? What is the problem of having a disease occuring twice a year, which is actually a common thing.
- line 171: is there a specific reason to choose 10%?

### Metrics
Maybe I am misunderstanding something but, if GTPA@k:
- "measures whether the differential diagnosis predicted by an agent contains the pathology p a patient was simulated from within its top-k entries"

and DDR@k:
- "measure whether the topk pathologies $\hat{Y}_{k}$ extracted from this distribution [distribution $\hat{Y}$ over the pathologies] are present in the topk pathologies Yk",

=> then isn't DDR@1 == GPTA@1?

### Minor remarks:
These in my opinion would help improving readability and ease comprehension:
- extensively define asap "antecedents"
- extensively define asap "Automatic Symptom Detection (ASD)" and "Automatic Diagnosis (AD)" underlying their differences
- line 35: if possible, split the references between ASD and AD rather then having a single long list
- line 97: HPO is an abbreviation but was never introduced.
- line 97: there should be a distinction between the reference for HPO (Köhler et al.) and the one for the ASD/AD system (Guan and Baral)

Supplementary material:
- Table 7,8,9 should be merged.

**Clarity:**

The paper is clearly written and the reader can follow the authors' arguments without effort.
The only minor defect is that equations are not numbered, which should be fixed.


**Correctness:**

To the best of my knowledge, my opinion is that the dataset is constructed in a sound way, however in my opinion certain aspects of the creation process necessitate further clarification.

**Documentation:**

The authors provide clear details w.r.t. the generation and organization of the dataset.
The dataset is freely available (CC BY 4.0 license) from a public URL and provides documentation w.r.t. to its content and intended uses.
Maintenance plan is not explicitly mentioned, though in my opinion in this case is not strictly necessary.

It must be noted that in the description found in the the dataset URL there is no disclaimer w.r.t to potential misuse.
I think this should be addressed since the dataset is aimed at applications in the biomedical domain.
The authors could include a sentence similar to the one they have in the paper:

"Finally, we would like to emphasize that this dataset should not be used to train
and deploy a model prior to performing rigorous evaluations of the model performance and verifying
that the system has proper coverage and representation of the population that it will interact with."

**Ethics:**

The dataset does not present ehtical concerns as it is synthetically generated.
The dataset is aimed at applications in the medical domain, but the authors specifically raise a warning in the paper w.r..t to its potential usage:  "Finally, we would like to emphasize that this dataset should not be used to train
and deploy a model prior to performing rigorous evaluations of the model performance and verifying
that the system has proper coverage and representation of the population that it will interact with."

**Relation To Prior Work:**

To the best of my knowledge all the relevant literature is correctly cited to have a clear understanding of the contribution and the position of the paper in the broader context of its research field.

**Summary And Contributions:**

The authors introduce a new dataset of synthetically generated data on patients for Automatic Symptom Detection (ASD) / Automatic Diagnosis (AD).

The major novelties (not in order of importance) of the dataset  are:
- being large-scale (1.3M patients)
- containing differential diagnosis, i.e. for each patient a list of plausible pathologies is provided
- introducing a distinction between "symptoms" and "antecedents" (patient history) and these being categorical and multi-choice rather than only binary.

In summary, to generate the data the authors make use of a propietary knowledge base, which provides provides probabilities for disease/symptoms/antecedents (conditioned on sex,age). These probabilities are then combined with census data and fed to a tool specifically developed to synthesize patients (Synthea) to generate the patient data. Subsequently the authors deploy a propietary ASD system to generate the differential diagnosis for each patient.

Finally the authors perform experiments with two existing (and tweaked) ASD/AD systems to asses the impact of training a system with the newly generated dataset.

---

> ### Author Response · Authors · 2022-08-19
> **Responses to questions of Reviewer USF5**
>
> Thank you for the feedback. Given that you had several questions, our response will be split into several openreview comments due to the 5000 character limit per comment.
>
> Please refer to the “General response to all reviewers” section for overall feedback to all reviewers.
>
> * W.r.t. (a) I cannot help but to notice that while the KB used to generated the data is adequately described in terms of its content, little to no information is provided as to "how" it was compiled. The only reference I could find was "which was constructed by compiling over 20 000 medical papers". Since this is a primary ingredient of the pipeline, I think that further details should be provided, e.g.: exactly who compiled it? Medical data can be "tricky" to interpret, was there any measure of the quality of the KB, e.g. in terms of agreement on diseases/sympots? What kind of medical papers? Where did they come from? I am not sure whether the nature of the resource prevents it, but answers to those questions seems to me mandatory. The same line of reasoning goes to for the "rule-based ASD" system used to generate the differential diagnosis. I think that the reader should be informed at least minimally as to how the system works, e.g. how are the different inputs processed and combined to arrive at the differential diagnosis?
>
> Following is feedback provided by one of the doctors who built the KB and the rule-based system: Doctors reviewed over 2 years of relevant papers on the diseases used to create the KB. The papers along with the medical experts' knowledge and experience were used to extract the typical and atypical presentations of diseases, along with the relevant symptoms and antecedent distributions to build accurate disease models based on demography and baseline patient characteristics. The process was exhaustive and independently validated by the doctors, where agreement was seeked for the presentation of every disease in the database. This is a strong point for the KB as it also uses the clinician experience to ensure that the diseases are accurately depicted across their usual and unusual presentations.
>
> The rule based system is a statistical engine that uses the patient response to generate a differential diagnosis in real-time. The engine has phases, where the first phase seeks to ask questions that have the highest probability of ruling out the most diseases in the initial differential. Subsequently, the engine seeks answers to questions linked to diseases in the ddx (differential) that represents the highest risk in terms of mortality and morbidity. Finally, the engine seeks to ask specific questions for the diseases in the remaining top-5 ddx and personal risk factors and antecedent.
>
> The engine was built for primary care and acute care setting, having in mind the goal of gathering a medical history that is as close as possible to the one clinicians would gather when evaluating a patient.
>
> The engine was tested on real patients in an acute care setting and the collected history was evaluated by doctors on a scale of pertinence and completeness.
> The ddx of the engine was compared to the clinician’s differential, who evaluated the patient with the usual clinical flow, blinded to the evidence collected by the engine initially.
>
> * W.r.t. (b), since the authors introduce a large-scale synthetic dataset, my expectation was that one evaluation experiment would have been devoted at cross-dataset validation, i.e. training a system on the synthetic dataset and then validate the gain in performance on a new unseen dataset. This would be a critical testbed considering the type of applications to which the dataset is aimed.
>
> We are attempting to respond to your comment but we’re not sure we fully understand it. If our response is incomplete, please let us know what you mean by “validate the gain in performance on a new unseen dataset”.
>
> The DDXPlus dataset contains a test set that can be used to evaluate models. Moreover, when creating the training, validation and test partitions, we did the split in a way that patient profiles (combination of age, sex, georaphic region, pathology, symptoms and antecedents, differential) are not shared between subsets.
>
> We cannot take a model trained on DDXPlus and evaluate it on other datasets because there are no publicly available datasets that contain the differential. Evaluating models on the ground truth pathology is an option rejected by the medical community as it is uninformative and not medically sound. Indeed, the evidence experienced by a patient will often lead to multiple possible pathologies (which constitute the differential), and this set of pathologies cannot be filtered without doing additional exams (testing, imagery, etc.). Finally, datasets do not have the same symptom definitions as those definitions can be at different levels of granularity and would require the help of doctors to establish a mapping; and many datasets do not contain antecedent information.

---

> > ### Author Response · Authors · 2022-08-19
> > **Additional responses to questions of Reviewer USF5**
> >
> > * Additionally, I think that experiment section would greatly benefit from an (even minimal) error analysis in order to assess the beneficial impact of the dataset. For instance, I was surprised to see that there wasn't a clear cut improvement in the "Positive evidence" (PE) - R/P/F1 metrics. Intuitively speaking, if a system is trained to predict a differential diagnosis (i.e. multiple diseases), shouldn't the coverage of the symptoms/antecedents be broader? While for AARLC system this seems to be the case (+ ∼ 20 in PER) it seems to have no effect for BASD.
> >
> > The idea behind these experiments was to show that it was possible to extend existing systems to predict the differential without sacrificing their ability to collect evidence. That being said, the ability of a system to collect positive evidence highly depends on how the system was designed. Based on the interactions we had with the doctor we are working with, we believe that most of the existing works do not design their solutions in a way that would be beneficial for the medical practitioners, and that would lead them to trust the results. This dataset introduces ingredients that could be leveraged to build appropriate systems, and thus paves the way for a better alignment between the ML and medical communities.
> >
> > Following is additional feedback provided by the doctor to your question: ”Finding the positive evidences paints an incomplete picture to the clinical team. As clinician we need to see that other evidences were seeked and ensure they were not present. This is very important for disease acuity, or less prevalent diseases. It is easy to find the positive evidence for common presentations, but as clinicians, we want to make sure other diagnosis were considered and were thoroughly evaluated. For example, if we take the case of a simple common cold. Only a few positive evidences will include the disease, but it's the broader negative evidence that covers the differential of similar yet very different in their management pathologies.”
> >
> > * The paper is clearly written and the reader can follow the authors' arguments without effort. The only minor defect is that equations are not numbered, which should be fixed.
> >
> > We fixed the equations. They are now numbered.
> >
> > *  It must be noted that in the description found in the the dataset URL there is no disclaimer w.r.t to potential misuse. I think this should be addressed since the dataset is aimed at applications in the biomedical domain. The authors could include a sentence similar to the one they have in the paper:
> >
> > Thank you for pointing this out. We added a disclaimer.
> >
> > * line 109: As a reader, I think I would like to know right away that the KB and the rule-based ASD system used to generate the data where provided by "Dialogue Health Technologies Inc" straight away, e.g. by having a footnote, rather finding this out in the acknowledgements.
> >
> > We made the change.
> >
> > * line 158-161: should mention at least a suggestion to overcome the limitation
> >
> > Following is feedback provided by the doctor working with us: “Using larger real world datasets would be the only way to fill those blanks. Unfortunately, some epidemiological data is lacking for some diseases as they don't carry any significance for public health, hence aren’t tracked. Using prevalence as a proxy to incidence is a reasonable alternative, especially if we target producing a good differential to cover the important questions to ask patients and make relevant clinical tools.”
> >
> > * line 164-168: if the KB was compiled by medical papers, the "domination" of specific pathologies is "reflecting" the true data distribution. Why change it? What is the problem of having a disease occuring twice a year, which is actually a common thing.
> >
> > You are right that the "domination" of specific pathologies is "reflecting" the true data distribution. That being said, we did not want a dataset that was dominated by 2-3 pathologies as this would have been less useful for the ML community. Rather, we wanted a more balanced dataset with the hope that ML models built on top of it would be efficient in handling every single pathology.
> >
> > * line 171: is there a specific reason to choose 10%?
> >
> > This was an arbitrary choice and the goal was to oversample rare but severe pathologies.

---

> > > ### Author Response · Authors · 2022-08-19
> > > **Final responses to questions of Reviewer USF5**
> > >
> > > * Maybe I am misunderstanding something but, if GTPA@k:
> > > "measures whether the differential diagnosis predicted by an agent contains the pathology p a patient was simulated from within its top-k entries"
> > > and DDR@k:
> > > "measure whether the topk pathologies
> > > Y^k
> > >  extracted from this distribution [distribution
> > > Y^
> > >  over the pathologies] are present in the topk pathologies Yk",
> > > => then isn't DDR@1 == GPTA@1?
> > >
> > >
> > > Those two metrics are different since the top pathology in the differential is not necessarily the ground pathology (cf Figure 4). Please note that the most important metrics are the ones measuring the quality of the differential. The GTPA metric is uninformative and not medically sound as doctors need to see a differential. We’re only reporting it here because other AD/ASD works have reported it but as stated in our overall response to the reviewers, we believe that those works are not focusing on the right problem definition (as specified by the medical community).

---

### Official Review · Reviewer_CUd3 · 2022-07-26
**Nice paper, I think that this paper introduces a valuable dataset for differential diagnosis**

**Rating:** 7
**Confidence:** 3
**Correctness:** I think the dataset is constructed in…
**Clarity:** The paper is well written.

**Strengths:**

- Clear difference with existing datasets. The limitations of existing datasets and the advantages of the proposed dataset are explained.
- Well description and organization of the released data. The paper includes a detailed data generation process and summary statistics, dataset documentation is also included in the Appendix.
- They reported the experimental results of the extended existing methods. The comparison results between differential diagnosis and not are analyzed to show the potential impact of the proposed dataset.

**Weaknesses:**

- Lacks the evaluation of medical experts for the assumptions and the generated data, although the paper explains the data are generated based on medical knowledge.
- The limitation of the proposed dataset can be described.

**Additional Feedback:**

See Weaknesses.

**Documentation:**

Data collection and organization are detailed in the paper and appendix. The data and code are available.

**Ethics:**

I have no other ethical concerns as this is a synthetic dataset.

**Relation To Prior Work:**

It is clearly discussed. The limitations of the existing dataset are described in Section 2, and the advantages of the proposed dataset are in Section 3.4.

**Summary And Contributions:**

This paper introduces a large-scale synthetic dataset focusing on cough, sore throat, and breathing problems for automatic medical diagnosis. They provide a good description of the information about the dataset. The experiments of two extended methods demonstrated that the collected dataset is useful for studying automatic diagnosis systems. Their contributions are:

- 1.3 million generated patients information covering 49 pathologies, 110 symptoms, and 113 antecedents.
- Well description and analysis of the released data.
- Methods, AARLC and BASD, are evaluated to show the usefulness of the collected dataset.

---

> ### Author Response · Authors · 2022-08-19
> **Responses to questions of Reviewer CUd3**
>
> Thank you for the feedback. Please refer to the “General response to all reviewers” section for overall feedback to all reviewers.
>
> * Lacks the evaluation of medical experts for the assumptions and the generated data, although the paper explains the data are generated based on medical knowledge.
>
> Following is the feedback provided by a doctor who is part of the team who created the KB and the rule-based system: “The medical knowledge base was constructed by doctors and domain experts using epidemiological data, clinical studies and domain expertise. It was independently reviewed and vetted by medical professionals and reflects the current medical knowledge for the presentation of the diseases.
>
> The ruled-based software uses the knowledge base. A statistical model calculated the probability of diseases based on the symptoms and past medical history (antecedents) of patients. Produced differential diagnosis by the engine was evaluated independently by three medical experts by comparing the expert’s differential diagnosis to the statistical engine. The software was tested with real life patients and the differential and gathered medical history was qualitatively assessed by a group of doctors.”
>
> The doctor also qualitatively evaluated the DDXPlus dataset and considered its quality to be good to build a proof of concept.
>
> *  The limitation of the proposed dataset can be described.
> We do not have the joint distribution between symptoms and antecedents and we mainly rely on marginal distributions assuming that in most of the cases, they were independent from each other. Those limitations, as well as other ones, are already described in the paper. Would you mind letting us know if our description is unclear or whether you’re looking for more information to be added?

---

### Official Review · Reviewer_UnxT · 2022-07-28
**Review for DDXPlus**

**Rating:** 5
**Confidence:** 3
**Correctness:** The claims appear correct.
**Clarity:** The paper is generally written well.

**Strengths:**

This paper has many strengths:
* The released dataset is extremely large
* The data include differential diagnosis information for synthetic patients, which is lacking in prior datasets
* Since the differential rules are probabilistic, it's believable that the synthetic data are generally realistic

**Weaknesses:**

This paper could be strengthened with the following considerations:
* Clearer discussion of what these data can be used for would be helpful. What would it take to adapt these data to a real case? Or if the data are primarily for evaluation, is there a benefit from the large size?
* The experiments only cover two existing systems, so it's not obvious how much room there is for improvement in existing systems according to these new data. Or if these two are representative, this should be explained further.
* The experimental results discussion should be expanded. How good should these metrics be? The F1 scores seem low, so it appears there's room for improvement on these data, but it would help to discuss the limitations of existing systems on these data to motivate how the community will use this new resource.

**Additional Feedback:**

Feel free to address any comments above

**Documentation:**

The documentation appears sufficient.

**Ethics:**

No ethical concerns.

**Relation To Prior Work:**

Since these data are so large, it would be helpful to list the typical scale of existing datasets, though the authors helpfully note that the prior works contain fewer features.

**Summary And Contributions:**

This work presents a new, large-scale synthetic dataset that can be used to evaluate differential diagnosis symptoms. By extracting diagnosis rules from medical literature, the authors automatically label a massive set of synthetic patients. Using these data, the authors compare two existing automatic diagnosis systems on their dataset, finding that incorporating differential diagnosis into the train improves their performance.

---

> ### Author Response · Authors · 2022-08-19
> **Responses to questions of Reviewer UnxT**
>
> Thank you for the feedback. Given that you had several questions, our response will be split into several openreview comments due to the 5000 character limit per comment.
>
> Please refer to the “General response to all reviewers” section for overall feedback to all reviewers.
>
> * Clearer discussion of what these data can be used for would be helpful. What would it take to adapt these data to a real case? Or if the data are primarily for evaluation, is there a benefit from the large size?
>
> Regarding the usefulness of this dataset, please refer to the “General response to all reviewers”.
>
> As to the question about the size of the dataset, we wanted to have a dataset that  was large enough to train RL models as many papers dealing with AD/ASD models use RL. Based on existing works that rely on SymCAT-based datasets, 1 million synthetic patients are generally used to train an RL agent.
>
> In terms of adaptation to a real case, we acknowledge that, being synthetic, our dataset is not perfect and may contain some unrealistic patients. However, as mentioned in the text, we encourage third parties prior to training and deploying a system based on this dataset to verify that the system has proper coverage and representation of the population that it will interact with. Moreover, we would like to  emphasize that before even considering applying models to real cases, we first need to make an effort in the ML community to work closely with the medical community to properly define the problem that needs to be solved. The current AD approaches published by the ML community with the goal of predicting the true pathology the patient is suffering from are not medically valid.
>
> Following is additional feedback from the doctor working with us: “For example, as a medical professional, a clinical tool that simply outputs a disease probability is not sufficient. When we evaluate a patient in an emergency setting, we want to explore diseases with a high mortality/morbidity even if the prior patient probability to have such disease is low, for obvious reasons. The dataset serves as a great baseline to build a trainable model, adapting a reward fonction to the clinical goal of the tool and working closely with medical professionals.”
>
> * The experiments only cover two existing systems, so it's not obvious how much room there is for improvement in existing systems according to these new data. Or if these two are representative, this should be explained further.
>
> The idea behind these experiments was to show that it is possible to extend existing systems to predict the differential without sacrificing their ability to collect evidence. Note that we recently identified further improvements to AARLC to better handle the differential. We ran new experiments and updated the results accordingly. There is significant room for improvement. Based on our work with the doctor and the explanations he provided as to the steps he follows when interacting with a patient to generate a differential, we can now better identify several design choices made in AARLC that we believe prevent this method from doing better. In a separate paper that we are working on, we are investigating a new method that can lead to much higher DDF1 and PER. The method is designed to try mimicking some of the reasoning steps used by doctors when interacting with patients. Initial results suggest that injecting this information in the model leads to much better performance.
>
> The DDXPlus dataset paves the way for the design of AD solutions that are more aligned with the needs of medical practitioners. For example, one could imagine a solution that:
> - Predicts the differential.
> - Collects evidence that supports the differential.
> - Encourages the collection of evidence that helps rule out severe pathologies that are not part of the differential, as this is something that doctors do. (Note that DDXPlus contains information about the severity of pathologies).
> - Makes sure that severe pathologies that are part of the ground truth differential are recovered in the predicted differential.
> We believe that existing approaches don’t do well because the currently available datasets do not contain the differential. Some datasets also don’t contain the pathology severity information. DDXPlus can help start to fill this gap. Moreover, the ML approaches will improve once ML researchers start working with the medical community to better define the problem.

---

> > ### Author Response · Authors · 2022-08-19
> > **Additional responses to questions of Reviewer UnxT**
> >
> > * The experimental results discussion should be expanded. How good should these metrics be? The F1 scores seem low, so it appears there's room for improvement on these data, but it would help to discuss the limitations of existing systems on these data to motivate how the community will use this new resource.
> >
> > As mentioned earlier, there is a disconnect between the way the ML community framed the AD problem and what people from the medical domain care about. Existing works focus on predicting the ground truth pathology with usually a minimum number of turns. Doctors do not reason in terms of a single pathology. They build a differential because the patient’s evidence isn’t always sufficient to identify the ground truth pathology. Moreover, doctors need to properly handle severe pathologies, to include the relevant ones in the differential and to explicitly exclude some of them from the differential. Finally, in order to trust ML models, doctors are looking for a sequence of questions that is medically reasonable. All of that indicates that the first step for improving performance is to build a problem definition with the medical community. We also need the help of this community to identify appropriate metrics to evaluate ML models. And we need datasets that can support the problem definition. DDXPlus was built with the help of a doctor after we initially tried to use a Symcat-based dataset and build an AD system that predicts the ground truth pathology. Our system had great performance on paper but when the doctor evaluated the interactions between our model and the Symcat-based synthetic patients, he rejected the model and indicated that its behavior wasn’t medically sound.
> >
> > All of that to say that results can be significantly improved if we have the right problem definition and datasets that support as much as possible this definition.
> >
> > *  Since these data are so large, it would be helpful to list the typical scale of existing datasets, though the authors helpfully note that the prior works contain fewer features.
> > Here are the statistics of the real patient datasets, Dxy and Muzhy:
> >
> >
> > |Statistics                      |    Dxy|      Muzhi|
> > |:------------------------------:|:-----:|:---------:|
> > |Number of pathologies           |      5|          4|
> > |Number of symptoms              |     41|         66|
> > |Number of cases in training set |    423|       568 |
> > |Number of cases in test set     |    104|         14|
> >
> >
> >
> > Note that none of those datasets have differentials. Moreover, in Dxy, the pathology associated with each medical diagnosis conversation was generated automatically by a medical dialogue system (instead of a human).
> >
> > Datasets generated from Symcat or other medical knowledge bases such as HPO can be  as large as we want them to be. Usually, previous works tend to do “on-the-fly” patient generation when using those knowledge bases to train their agents. The training sets can contain up to 1 million synthetic patients.

---

### Official Review · Reviewer_GwmY · 2022-07-28
**Review of A New Dataset For Automatic Medical Diagnosis**

**Rating:** 8
**Confidence:** 3
**Correctness:** The claims appear to be sound and rea…
**Clarity:** The paper is well motivated and easy …

**Strengths:**

- The paper is well-motivated and differentiates itself from existing works.
- The dataset contains a large scale of data that includes various pathologies, symptoms, and antecedents.
- The dataset contains not only binary but also categorical and multi-choice symptoms and antecedents.
- The data generation process seems well-thought-out.

**Weaknesses:**

- It would be great to have some sample data instance either in the main paper or the supplementary material.

**Additional Feedback:**

N/A

**Documentation:**

Sufficient documentation has been provided.

**Relation To Prior Work:**

Related works are well discussed.

**Summary And Contributions:**

The authors propose a large-scale synthetic dataset of differential diagnosis, ranking a list of plausible diseases that a patient might suffer. Unlike the previous datasets, the proposed dataset contains more pathologies, symptoms, and antecedents with non-binary questions and differential diagnosis data. The potential usage of the dataset is to assist doctors in telemedicine service.

---

> ### Author Response · Authors · 2022-08-19
> **Responses to questions of Reviewer GwmY**
>
> Thank you for the feedback. Please refer to the “General response to all reviewers” section for overall feedback to all reviewers.
>
> * It would be great to have some sample data instance either in the main paper or the supplementary material.
>
> We added some sample data instances in the main text (Section 3.5) and in the supplementary material (Section A.9). We also added an analysis done by the doctor of the predictions made by the models for one patient in Section A.10.

---

### Official Review · Reviewer_NKAP · 2022-07-28

**Rating:** 4
**Confidence:** 5

**Strengths:**

The paper has following strengths:
- Although there are similar datasets, DDXPlus is the only one that comes with a paired differential diagnosis.

- DDXPlus clearly divides evidences into to subsets (i.e. symptoms and antecedents), which could potentially be useful for other researchers.

- The dataset construction process is clearly described.


**Weaknesses:**

The paper has following weaknesses.
- The paper claims that, unlike previous datasets that only use binary symptoms, DDXPlus uses multi-class and multi-label symptoms. However, this can also be achieved with previous datasets by post-processing. Based on simple domain knowledge, individual evidences can be grouped into a higher-level evidence (e.g. chest pain, neck pain, wrist pain, ... ==> pain), which will convert a subset of evidences from the existing datasets to multi-label/multi-class evidences.

- The paper claims that it is a large-scale dataset, unlike existing datasets. However, some existing datasets (e.g. SymCAT, HPO) are theoretically unbounded in size, because a user can generate as many records as they need, because they geneyrate synthetic records by exploiting pre-defined domain knowledge (e.g. medical ontology, census data). This is actually the same for DDXPlus, which also generates purely synthetic records based on on medical ontology and census data. DDXPlus could have been a billion-size dataset if the authors chose so, and there is no justification as to why 1.3 million records were generated as the final dataset.

- Although DDXPlus is the only dataset with the ground-truth (GT) differential diagnosis (ddx), they are generated by a rule-based telemedicine software. Therefore the quality of the GT ddx is in serious question. Note that an automatic diagnosis model (e.g. AARLC) can learn to generate ddx by training on previous datasets (e.g. SymCAT, HPO, Muzhi) that do not have GT ddx. The limitation is that there is no way to evaluate if the model-generated ddx is clinically meaningful. DDXPlus can potentially fill this gap, because it claims to have GT ddx. But since that GT ddx is generated by a very questionable rule-based software (which cannot be open-sourced because it is a proprietary product), there is hardly any value in it. Moreover, if ddx can be generated by a rule-based process, why should we train machine learning models to do it?

- Although the authors claim that training the model to predict ddx instead of the final pathology is helpful, the experimental results say otherwise. According to BASD's performance in Table 1, using ddx during training only degrades the prediction performance of the true pathology, and makes negligible improvement in the model's ability to find the evidences.

Overall, it is very difficult to consider DDXPlus, which is a synthetic derivative of a census data, medical knowledge base (the quality of which is unknown) and a rule-based system (also quality unknown), a new dataset that brings significant new value to the community.


**Additional Feedback:**

Is there any future plan to open-source the proprietary medical knowledge base and the telemedicine software? It would be way easier for the research community to study them to learn new insight, rather than try to replicate their functionalities by using their synthetic derivativess.

**Clarity:**

The paper is overall nicely structured and very well-written.
All the necessary details that are required to understand the dataset construction process is provided in a detailed manner.
Especially, the elucidation of the assumptions made during the synthesis of patient records is much appreciated.

**Correctness:**

Although the overall dataset construction process is reasonable, it has several shortcomings due to the fact that the dataset is completely synthetic, and completely relies on existing medical ontology and a rule-based software. The details of the shortcomings are described in the Weaknesses section.

**Documentation:**

The proposed dataset is well-documented.

**Ethics:**

Since the dataset is completely synthetic, there is no ethical concern.

**Relation To Prior Work:**

The paper makes several claims as to how DDXPlus is different from the datasets, such as the size, types of symptoms, having differential diagnosis, etc. Although a couple of them are true, the others do not seem as a solid difference. The details are described in the Strenghts and Weaknesses section.

**Summary And Contributions:**

This work proposes DDXPlus, a symptom-disease dataset of synthetic patient records containing demographic information (age, gender, geographic), each patient's disease, symptoms, antecedents and differential diagnosis.
Each patient's disease and evidences (i.e. symptoms and antecedents) are sythetically generated based on a proprietary KB and a commercial synthesizer (i.e. Synthea).
Differential diagnosis for each patient is generated by feeding all evidences to a commercial telemedicine software, which generates a set of possible diseases.
DDXPlus differs from previous datasets in several ways such as providing a differential diagnosis, size of the dataset, using symptoms that are non-binary, etc.
In the experiments, existing models that can perform automatic diagnosis was applied to DDXPlus and demonstrated good performance.

---

> ### Author Response · Authors · 2022-08-19
> **Responses to questions of Reviewer NKAP**
>
> Thank you for the feedback. Given that you had several questions, our response will be split into several openreview comments due to the 5000 character limit per comment.
> Please refer to the “General response to all reviewers” section for overall feedback to all reviewers.
>
> * The paper claims that, unlike previous datasets that only use binary symptoms, DDXPlus uses multi-class and multi-label symptoms. However, this can also be achieved with previous datasets by post-processing. Based on simple domain knowledge, individual evidences can be grouped into a higher-level evidence (e.g. chest pain, neck pain, wrist pain, ... ==> pain), which will convert a subset of evidences from the existing datasets to multi-label/multi-class evidences.
>
> It is true that other datasets can be post-processed to support multi-choice and categorical evidence. But deciding which evidence to combine and how to combine it is not trivial, as the evidence needs to be organized in a way that facilitates the interaction between the doctor and the patient in order to collect clear and comprehensive information from the patient. Following is the feedback sent by the doctor working with us and who contributed to the creation of the medical knowledge and the categorization of evidence into binary, categorical and multi-choice types: “The DDXPlus dataset has a major advantage over other datasets for multi-class and multi-label symptoms. Since almost 60% of all presentations in real life will include “pain”, a great level of attention was given to the description of pain for each disease where this symptom can be found. This feature includes the localization, radiation, intensity, subjective characterization of common pain description by patients, a precision feature (very small area to diffuse) and a rapidity of onset. Just the pain symptom encompasses all these sub-features and was created using domain experts and medical journal articles looking at disease presentations. This would be extremely hard to derive from another dataset, since we have not seen other ones cover the pain description extensively, although one of the most important symptoms a clinician will spend a good amount of time defining clearly with the patient.
> Same for skin rashes, the rash description includes the usual dermatological lesions characterization used by clinicians when evaluating a patient.”
>
> * The paper claims that it is a large-scale dataset, unlike existing datasets. However, some existing datasets (e.g. SymCAT, HPO) are theoretically unbounded in size, because a user can generate as many records as they need, because they geneyrate synthetic records by exploiting pre-defined domain knowledge (e.g. medical ontology, census data). This is actually the same for DDXPlus, which also generates purely synthetic records based on on medical ontology and census data. DDXPlus could have been a billion-size dataset if the authors chose so, and there is no justification as to why 1.3 million records were generated as the final dataset.
>
> It is true that we can generate a very large number of synthetic patient records using existing medical knowledge bases such as Symcat and HPO. But the resulting datasets do not contain the differential diagnosis information. Those medical knowledge bases can only be used to synthesize patients with an underlying pathology but cannot simulate the differential diagnosis that a doctor generates when examining a patient. It is important to keep in mind that doctors do not reason in terms of a single underlying pathology; they instead reason in terms of a differential diagnosis and expect automatic diagnosis systems to do the same.
>
> Moreover, previous works relying on the Symcat and HPO knowledge bases tend to generate patients on the fly during training and evaluation. While the “on-the-fly patient generation” process is flexible, we believe that having predefined sets of patients (such as in DDXPlus) provides non-negligible advantages: 1) There is a predefined test set to compare different methods. This could have been done with Symcat and HPO but no one has done it so far. 2) The “on-the-fly patient generation” doesn’t guarantee that rare (but sometimes life-endangering) pathologies appear in the “on-the-fly” test sets. In DDXPlus, we used stratified sampling to ensure that all pathologies are represented in the train, validation and test partitions. 3) The doctor with whom we’ve been working indicated that he has more confidence in a static dataset because this allows him to properly analyze the patients (within and across the train, valid, and test partitions) to better analyze the behavior of a model that uses those patients.
>
> As for the number of patients in DDXPlus, we believe that ~1 million patients in the training set to be a reasonable target based on the number of samples used to train RL (reinforcement learning) agents in previous works. We did not attempt to generate a larger dataset although this would be possible.

---

> > ### Author Response · Authors · 2022-08-19
> > **Additional responses to  questions of Reviewer NKAP**
> >
> > * Although DDXPlus is the only dataset with the ground-truth (GT) differential diagnosis (ddx), they are generated by a rule-based telemedicine software. Therefore the quality of the GT ddx is in serious question. Note that an automatic diagnosis model (e.g. AARLC) can learn to generate ddx by training on previous datasets (e.g. SymCAT, HPO, Muzhi) that do not have GT ddx. The limitation is that there is no way to evaluate if the model-generated ddx is clinically meaningful. DDXPlus can potentially fill this gap, because it claims to have GT ddx. But since that GT ddx is generated by a very questionable rule-based software (which cannot be open-sourced because it is a proprietary product), there is hardly any value in it. Moreover, if ddx can be generated by a rule-based process, why should we train machine learning models to do it?
> >
> > There are 3 topics listed in the weakness: 1) quality of the rule-based system used to generate the differential, 2) the notion that AD systems can generate a differential based on previous datasets, 3) whether machine learning models need to be trained to generate a differential.
> >
> > Topic #1: Following is the feedback provided by a doctor who is part of the team who created the KB and the rule-based system: “The ruled-based software uses a knowledge base constructed by doctors and domain experts using epidemiological data, clinical studies and domain expertise. A statistical model calculated the probability of diseases based on the symptoms and past medical history (antecedents) of patients. Produced differential diagnosis by the engine was evaluated independently by three medical experts by comparing the expert’s differential diagnosis to the statistical engine. The software was tested with real life patients and the differential and gathered medical history was qualitatively assessed by a group of doctors. Although the proprietary software performed well, the non-trainable aspect was found to be a limiter for future improvement and building a trainable agent seemed advantageous.”
> >
> > We would also like to point out that the doctor made a qualitative analysis of the DDXPlus dataset by analyzing a subset of patients and found the quality to be good.
> >
> > Topic #2: Regarding the assertion according to which an automatic diagnosis model (e.g., AARLC) can learn to generate the ddx by training on previous datasets (e.g., SymCAT, HPO, Muzhi), we would like to emphasize that any model which predicts a posterior distribution over the set of pathologies could argue that that distribution is the resulting ddx. However, as shown by the results presented in Table 4, models that are just trained using the GT pathology are not able to recover the GT ddx. Those models have a high DDP but a low DDR, because nearly all the probability mass is assigned to a single pathology and the other pathologies get very low probabilities and are discarded. This is a show-stopper for doctors. Doctors want to be reassured that a model is capable of properly identifying the short list of pathologies the patient might be suffering from based on the evidence collected from the patient and they want to see that the probabilities of those pathologies are not very small. We need to keep in mind that follow-up medical exams (such as X-rays, MRIs, etc.) can be harmful to patients and very expensive. It is therefore crucial to identify the right differential and ask only for the necessary follow-ups.
> > We would also like to provide the following feedback from the doctor working with us: “One weakness of non GT ddx trained models is that although the models can perform well on paper when looking at the evaluation metrics, the collected question/answer pairs don’t elicit a high level of confidence by the clinicians as the underlying reasoning seems far off the usual one. We found that using a GT ddx, the doctors could understand the process behind the choice of questions and lead to a medical history that had an actual utility in practice.”
> >
> > Topic #3: When we used the rule-based system to generate the differential, we fed all the patient evidence at once to the system to make sure that the system could use all the available information to generate the differential. But in practice, an ASD or AD system has to figure out which evidence to ask about. Those systems cannot replicate the approach we used to generate the differential as they are not fed the patient evidence. Instead, they are responsible for collecting this evidence and then generating the differential. If they have a poor inquiry policy, they will generate poor differentials. Moreover, generating the differential is not the only goal of such systems. To be accepted by doctors, those systems must generate the differential, ask relevant and sufficient questions about evidence, and properly handle severe pathologies (to explicitly rule them in or out of the differential).

---

> > > ### Author Response · Authors · 2022-08-19
> > > **Final responses to questions of Reviewer NKAP**
> > >
> > > * Although the authors claim that training the model to predict ddx instead of the final pathology is helpful, the experimental results say otherwise. According to BASD's performance in Table 1, using ddx during training only degrades the prediction performance of the true pathology, and makes negligible improvement in the model's ability to find the evidences.
> > >
> > > Predicting the true pathology from the patient evidence is not a medically sound approach, as the evidence is sometimes insufficient to pinpoint the single pathology the patient is suffering from and medical follow-ups are required to identify this pathology. Doctors generate a differential, which is a short list of pathologies that is derived from the patient's evidence. To provide a simple example based on the recent covid pandemic: imagine a patient experiencing fever and cough. Based on those symptoms alone, it is impossible to determine if the patient is suffering from covid, the flu, pneumonia or another relevant disease. Testing is required. For example, the doctor might ask the patient to do a covid test. A model that predicts a single pathology based on this patient’s evidence is a poor model that will be categorically rejected by the medical community. This is why the GTPA@k metric is not meaningful. We are presenting it simply because this is the metric used in other AD/ASD papers that do not predict the differential. The metrics that matter are DDF1 and PER. There are additional metrics that matter and haven’t been introduced in this paper: those are metrics related to measuring a model’s ability to properly handle severe pathologies and metrics to quantify whether the sequence of questions asked by a model follows the approach used by doctors. We are working on proposing such metrics based on our collaboration with doctors.
> > >
> > > As for the degradation of the GTPA@k when using the ddx, this is to be expected, in particular for GTPA@1, because the GT pathology is not always the most likely pathology in the differential, as shown in Figures 12, 14 and 16. We would like to reiterate that predicting the ground truth pathology is not a medically sound approach. This is the approach that has been used in the AD/ASD papers published by the ML community but those papers are not solving the problem as defined by the medical community. It is paramount that we work with the medical community to properly define the problem and build useful models.
> > >
> > > Following is additional feedback provided by the doctor about the GTPA@k being worse when predicting the ddx: “This is expected. In clinical practice, the GT pathology cannot often be verified or is not known initially. For example, patients will be admitted based on suspicion of diseases from a differential produced by a first doctor. Then during the admission, the GT pathology will be discovered from testing, imagery, etc... The ddx is then the only thing of importance at the beginning of the evaluation as it allows to plan further testing and care planning. In other words, even if some evidence is not found, the pertinent negatives and the suspected differential have more value in practice when the presentation of disease is atypical.”
> > >
> > > * Overall, it is very difficult to consider DDXPlus, which is a synthetic derivative of a census data, medical knowledge base (the quality of which is unknown) and a rule-based system (also quality unknown), a new dataset that brings significant new value to the community.
> > >
> > > Following is feedback from the doctor  who was part of the team who created the KB and the rule-based system: “The knowledge base was created by field experts, mainly by using the latest epidemiological data available in clinical studies. The clinical study database was independently verified by doctors in the context of a due diligence for acquisition by Dialogue Technologies. To our knowledge, the dataset is the only one that included a complete pain assessment and multi class multi label symptoms.”
> > >
> > > The rule-based system was also qualitatively evaluated by doctors as described in one of our aforementioned responses.
> > >
> > > Moreover, the dataset is providing differentials and information about severe pathologies and can help pave the way for an improved problem definition as the problem definition that the ML community is currently working on is incomplete and incorrect.
> > >
> > > * Is there any future plan to open-source the proprietary medical knowledge base and the telemedicine software? It would be way easier for the research community to study them to learn new insight, rather than try to replicate their functionalities by using their synthetic derivativess.
> > >
> > > No, there aren’t future plans for Dialogue Technologies to open-source the KB and the telemedicine software.

---

### Author Response · Authors · 2022-08-19
**General response to all reviewers**

Based on the reviewers' questions, we would like to explain the motivation behind this dataset. This work is part of a project where we were tasked to build an AD system. We initially trained an RL agent which mimicked the techniques proposed in papers published by the ML community, and whose goals are to predict the patient’s underlying pathology while asking as few questions as possible. This agent was trained using a synthetic dataset derived from the SymCat medical knowledge base, and had a high GTPA and a low IL. We therefore assumed that our model was very good. At that point, we asked a doctor to qualitatively analyze the performance of our agent. The doctor verified interactions between the agent and synthetic patients and was very disappointed. He indicated that the agent was “cutting corners” by asking too few questions, often leading to an incomplete medical reasoning, and he said that forcing the agent to identify the patient’s pathology didn’t make sense as the evidence provided by patients is often insufficient to identify the underlying pathology. He said that for him to trust such a tool, the tool would need to consider important aspects of a doctor’s reasoning. Those include: 1) Generating a differential diagnosis, as it is sometimes impossible to identify the pathology the patient is suffering from just by inquiring about the patient’s symptoms/antecedents. 2) A doctor needs to pay special attention to severe pathologies as a mistake in handling such pathologies can lead to major repercussions for patients. 3) A doctor needs to make sure to collect sufficient and relevant evidence. With this feedback, it became clear to us that there is a disconnect between the AD problem as defined by the ML community and what the medical community cares about.

Let’s look at a patient from a medical provider’s perspective:


    Sex: F, Age: 79
    Geographical region: North America
    Pathology: Spontaneous pneumothorax
    Symptoms:
    ---------
	 - I have chest pain even at rest.
	 - I feel pain.
	 - The pain is:
		 » a knife stroke
	 - The pain locations are:
		 » upper chest
		 » breast(R)
		 » breast(L)
	 - On a scale of 0-10, the pain intensity is 7
	 - On a scale of 0-10, the pain's location precision is 4
	 - On a scale of 0-10, the pace at which the pain appear is 9
	 - I have symptoms that increase with physical exertion but alleviate with rest.
    Antecedents:
    -----------
	 - I have had a spontaneous pneumothorax.
	 - I smoke cigarettes.
	 - I have a chronic obstructive pulmonary disease.
	 - Some family members have had a pneumothorax.
    Differential diagnosis:
    ----------------------
    Unstable angina: 0.262, Stable angina: 0.201, Possible NSTEMI / STEMI: 0.160, GERD: 0.145, Pericarditis: 0.091, Atrial fibrillation: 0.082, Spontaneous pneumothorax: 0.060


The diagnosis is a spontaneous pneumothorax, which is atypical for this patient’s demography. It is rare, but possible. We see the patient’s positive symptoms and the produced differential. If we were only to collect the positive evidence and score solely on the true or false label, as a doctor, I would find the history missing key questions for a 79 yo female presenting with chest pain that increases with exertion. We definitely need to cover the cardiovascular review more thoroughly and explore the most specific symptoms and risk factors. Optimizing for an accurate differential forces the agent to ask questions that in that case would be answered “no”, but would have a really positive impact on the confidence of doctors and medical staff in the agent's capability to adequately collect a medical history. Without a differential, much would be left out of the medical history.


We spent time with the doctor to better understand the problem definition and we started thinking about models that could be built based on this definition. But we realized that none of the existing datasets contained the differential diagnosis and pathology severity information. We therefore worked with the doctor to generate the DDXPlus dataset. Once generated, we asked the doctor to do a qualitative evaluation of the synthetic patients. He indicated that the quality was good (but obviously not perfect) and felt comfortable to have this dataset be used to build a proof of concept.

By making this dataset publicly available, we’re hoping to help the ML community focus on a problem definition that makes sense to the medical community as the current models will not be accepted by the medical community.

Finally, we would like to emphasize that we are not advocating using DDXPlus to deploy a model. As stated above, before even contemplating releasing a model, we first need to make sure the ML community is working on solving the right problem.

---

> ### Author Response · Authors · 2022-08-19
> **Additional information to all reviewers**
>
> Side notes:
> 1) We recently further improved the AARLC baseline and updated the results of Tables 3 and 4.
> 2) We trained each model 3 times and reported mean performance with 95% confidence intervals.
> 3) We refined the metrics to stop replicating metrics used in other AD/ASD papers and which are not medically useful: we removed PEP and PEF1 as it is ok to ask negative questions (which is something that doctors do).
> 4) We added sample patients in sections 3.5 and A.9. The doctor commented 3 of them. We also added the doctor's evaluation of the predictions made by the 4 models presented in Table 3 for the sample patient presented in section 3.5.

---

### Meta-Review · Area_Chair_8hGs · 2022-09-07

**Recommendation:** Accept
**Confidence:** 3

**Metareview:**

This paper proposes a symptom-disease dataset of synthetic patient records containing demographic information, disease, symptoms, antecedents and differential diagnosis. The disease, evidences, and differential diagnoses are synthetically generated which makes the dataset easy to share. Several reviewers pointed out important concerns in the methods used to synthesize the data which may raise ethical problems if it is assumed that performance on this dataset is indicative of performance on real patients. Nevertheless, I believe that this dataset can be of value to the community as it can help the community develop better ways to connect differential diagnosis to the AD/ASD task (while we wait for a dataset with real differential diagnoses). I urge the authors to clearly caution readers about the datasets limitations!

---

### Decision · Program_Chairs · 2022-09-16

Accept